# Inhibition of Mitochondrial Antioxidant Defense and CDK4/6 in Mesothelioma

**DOI:** 10.3390/molecules28114380

**Published:** 2023-05-27

**Authors:** Marian Kratzke, George Scaria, Stephen Porter, Betsy Kren, Mark A. Klein

**Affiliations:** 1Research Service, Minneapolis VA Health Care System, Minneapolis, MN 55417, USA; 2Hematology/Oncology Section, Primary Care Service Line, Minneapolis VA Health Care System, Minneapolis, MN 55417, USA; 3Division of Hematology, Oncology and Transplantation, Department of Medicine, University of Minnesota, Minneapolis, MN 55417, USA; 4Department of Laboratory Medicine and Pathology, University of Minnesota, Minneapolis, MN 55417, USA

**Keywords:** mitochondria, antioxidant, cell cycle, cyclin-dependent kinase, cyclin, mesothelioma

## Abstract

Advanced mesothelioma is considered an incurable disease and new treatment strategies are needed. Previous studies have demonstrated that mitochondrial antioxidant defense proteins and the cell cycle may contribute to mesothelioma growth, and that the inhibition of these pathways may be effective against this cancer. We demonstrated that the antioxidant defense inhibitor auranofin and the cyclin-dependent kinase 4/6 inhibitor palbociclib could decrease mesothelioma cell proliferation alone or in combination. In addition, we determined the effects of these compounds on colony growth, cell cycle progression, and the expression of key antioxidant defense and cell cycle proteins. Auranofin and palbociclib were effective in decreasing cell growth and inhibiting the above-described activity across all assays. Further study of this drug combination will elucidate the contribution of these pathways to mesothelioma activity and may reveal a new treatment strategy.

## 1. Introduction

Mesothelioma is a devastating and recalcitrant cancer that arises from the pleura or lining of the peritoneum. Treatments for advanced mesothelioma in the first-line setting may include chemotherapy with cisplatin and pemetrexed or immunotherapy with checkpoint inhibitors, primarily nivolumab and ipilimumab [1,2]. However, mesothelioma is incurable in the majority of cases with improvement in median overall survival with chemotherapy or immunotherapy measured in months [1,2]. To date, clear mechanisms for lack of longer term benefit from current treatments have been elusive. In addition, biomarkers for treatment of mesothelioma have not yet been validated. New treatment strategies against mesothelioma are needed. Two pathways that show some promise but remain to be significantly studied include mitochondrial antioxidant defense and the cell cycle.

### 1.1. Mitochondrial Antioxidant Defense

Two antioxidant pathways consisting of (1) [glutathione (GSH)/glutathione peroxidase (GPx)/glutathione-S-transferase (GST)/glutaredoxin 2 (Grx2)] and (2) [thioredoxin 2 (Trx2)/peroxiredoxin 3 (Prx3)/methionine sulfoxide reductase] are present in the mitochondria and, in combination with protein thiols chemistry, contribute substantially to the processing of intracellular reactive oxygen species (ROS) that occur due to energy generation or hypoxia-induced stress [3]. Trx2 contributes to cell and mitochondrial viability and to the response to oxidative stress [4,5]. Trx2 haploinsufficient (Trx2 +/-) mice show impaired mitochondria function and oxidative stress, decreased ATP production, and increased oxidative damage to nuclear DNA, lipids, and proteins [6]. TNF-α-induced ROS generation, NF-κB activation [7], mitochondrial permeability transition (mPT) [8], and apoptosis [9] are regulated by Trx2. Finally, overexpression of Trx2 is associated with increased mitochondrial membrane potential (∆ψm) [10].

Mitochondrial antioxidant defense proteins play a key role in contributing to the ability of mesothelioma cells to counteract pro-apoptotic signaling [11,12,13,14,15,16]. Increased expression of peroxiredoxin 3 (Prx3) and manganese superoxide dismutase (MnSOD) has been demonstrated in mesothelioma tumors in clinical studies [14,15]. MnSOD activity is increased after phosphorylation by CDK4, causing superoxide levels to decline in human keratinocytes. Downregulation of Trx2 and inhibition of Prx3 result in increased ROS production in mesothelioma cells and decreased mesothelioma tumor growth. FOXM1-transcribed Cyclin D1 and CDK4 are associated with increased activity of this pathway [17,18]. Auranofin has previously been demonstrated to downregulate Trx2 and has been effective against cancer cells in preclinical studies [11,12,13,19].

### 1.2. Cell Cycle-Related Proteins

Cyclin-dependent kinases 4 and 6 (CDK4/6) are major proteins responsible for progression through G1 to S phase, and regulation of this step is altered in many cancers [20,21,22,23]. Cyclin D1 binds to CDK4 or CDK6 and these complexes promote hyper-phosphorylation of retinoblastoma protein (Rb). The tumor suppressor and CDK inhibitor p16INK4a (p16) inhibits the CDK4/Cyclin D or CDK6/Cyclin D [20,24]. Several studies have demonstrated that p16 loss (usually via deletion or via methylation) is common in mesothelioma [25]. Deletion of the 9p21 locus that encodes p16 was deleted in 35/40 cases (88%) in one study [25]. Loss of p16INK4a/p14ARF was identified in 49/71 (70%) epithelioid tumors, 16/19 biphasic (89%) tumors, and 5/5 (100%) sarcomatoid tumors in a different study [25]. 

Palbociclib is a CDK4/6 inhibitor that inhibits growth of Rb-intact breast cancer xenografts [26]. The clinical benefit of palbociclib was established in the PALOMA-1 trial, where participants with breast cancer were randomized to letrozole + palbociclib versus letrozole alone [27]. Progression-free survival was (PFS) 10 months in the letrozole arm and 20 months in the combination arm. Nearly all mesothelioma tumors will exhibit deregulated G1/S cell cycle control (Rb-intact), but only a few attempts have been made to evaluate CDK4/6 inhibition in mesothelioma. For example, Bonelli et al. have evaluated palbociclib previously in two studies, demonstrating some potency alone or in combination [28,29].

### 1.3. Interaction between Mitochondrial Antioxidant Defense and the Cell Cycle

Increased expression of peroxiredoxin 3 (Prx3) and manganese superoxide dismutase (MnSOD) has been demonstrated in mesothelioma tumors [14,15]. CDK4 has been demonstrated to increase MnSOD activity via phosphorylation, causing superoxide levels to decline in keratinocytes [18]. FOXM1-transcribed Cyclin D1 and CDK4 enhance activity of this pathway [17,18]. Based on these findings that potentially connect mitochondrial antioxidant defense and cell cycle pathways, we hypothesized that downregulation of Trx2 via auranofin combined with CDK4/6 inhibition via palbociclib would be more effective than either agent alone to decrease mesothelioma cell growth. In this paper, we describe in vitro activity of either agent alone or in combination against mesothelioma cells.

## 2. Results

### 2.1. Inhibition of Cell Proliferation by Auranofin and Palbociclib

The growth-inhibitory effects of CDK4/6 inhibition were evaluated by determining the IC_50_ for palbociclib and auranofin against mesothelioma cell lines H2052, H2373, H2452, and H2461. Met5A cells were utilized as a non-transformed control cell line. Palbociclib and auranofin both resulted in decreased cell proliferation at 72 h for all cell lines (Figure 1A,B). 

For comparison to a current standard of care treatment in mesothelioma, H2373 and H2452 cells were exposed to increasing amounts of pemetrexed, a folate pathway inhibitor in similar fashion to palbociclib and auranofin. Figure 2 shows the IC_50_ curve for pemetrexed in comparison to palbociclib and auranofin. The IC_50_ was 14.58 nM for pemetrexed in H2373 cells and 34.71 nM in H2452 cells. We also exposed control 3T3 cells to palbociclib and auranofin in separate experiments for 72 h to determine effects on a control line. The IC_50_ was 2.71 μM for auranofin and 22.6 μM for palbociclib.

### 2.2. Inhibition of Cell Proliferation by Combination of Auranofin and Palbociclib

We hypothesized auranofin and palbociclib could act in an additive fashion or synergistic fashion; therefore, we conducted cell proliferation experiments with varying combinations of concentrations of each agent per the methods of Chou–Talalay [30]. Figure 2 exhibits viability relative to control for each drug concentration combination for cell lines H2373 and H2452. For both cell lines, several combinations were consistent with a synergistic effect in decreasing cell proliferation (Figure 3 and Figure 4). Table 1 exhibits the combination indices for each combination of palbociclib and auranofin. Numerous combinations indices were < 1, suggesting synergism.

### 2.3. Colony-Forming Assays

Palbociclib and auranofin each significantly decreased H2373 and H2452 colony formation (Figure 5). Palbociclib resulted in nearly complete inhibition of colony formation at a concentration of 50 μM in both cell lines. Auranofin resulted in nearly complete inhibition of colony formation at 5 μM and at 10 μM. These concentrations were very close to the relative IC_50_ values for each drug against mesothelioma cell lines as in Figure 1. 

### 2.4. Cell Cycle Analysis

Effects of palbociclib on the cell cycle following a 48 h treatment in mesothelioma cells demonstrated distinct inhibition of the cell cycle at G1/S with a significant increase in the percentage of cells in G1 as evaluated by flow cytometry. Treatment with auranofin in mesothelioma cell lines resulted in an increased proportion of cells in the sub-G1 fraction which may indicate an apoptotic response to the thioredoxin inhibitor. See Figure 6 for results for both experiments.

### 2.5. Modulation of Gene Expression in Response to Treatment with Palbociclib or Auranofin

We examined the expression of the four predominant cell cycle cyclins and CDKs to establish if inhibition of CDK4/6 by palbociclib was associated with alteration of CDKs or cyclins that have been implicated in acquisition of resistance to palbociclib in luminal breast cancer [31]. Substantially enhanced expression of Cyclins D1 and E1 were observed in H2052, but not in either H2373 or H2452. In contrast, auranofin resulted in the loss of Cyclin D1 protein expression in two of the three cell lines but did affect Cyclin E1 expression (Figure 7). Both drugs resulted in the loss of Cyclin A and B1 gene expression by 72 h (Figure 7), although the loss was less pronounced in the H2452 cells being observed only at the higher concentrations employed for both drugs. Auranofin at 2.5 μM for 72 h was > 90% lethal for the two Bap1-expressing cell lines H2052 and H2373 and thus was not analyzed.

Palbociclib led to a substantial loss of CDK1 gene expression in all three mesothelioma cell lines. Auranofin treatment was variably associated with decreasing CDK1 levels, depending on the cell line and drug concentration which mirrored the pattern observed for Cyclin D1 loss (Figure 7). CDK2 levels increased in all palbociclib-treated cells, whereas CDK4 and CDK6 levels were only modestly increased or unchanged with palbociclib treatment. Auranofin resulted in loss of CDK2, CDK4, and CDK6 expression in two of the three cells lines (Figure 7, Appendix A).

We examined both FOXM1 and Rb transcription factors as both are known targets for the Cyclin D1/CDK4 phosphorylation that promotes cell cycle progression. As shown in Figure 8, treatment with palbociclib substantially reduced the phosphorylation of Rb at S780, a site acted on equally by both Cyclin D1/CDK4 and Cyclin D1/CDK6. The auranofin-treated cells had a much smaller loss of pRb S780 expression, but in contrast to the palbociclib treatment, overall levels of Rb expression decreased as well, rather than the shift from the phosphorylated to the non-phosphorylated form of Rb observed with palbociclib. FOXM1 expression was lost in response to either drug treatment, with palbociclib inducing a greater loss. 

Cell cycle arrest is effective in reducing the rate of tumor growth; however, treatment-induced cell death is required for tumor shrinkage and thus we examined the expression of pro-apoptotic and pro-survival genes whose levels are modulated in malignant mesothelioma [32,33,34]. The pro-apoptotic genes Bax and/or Bid were induced in response to both auranofin or palbociclib, although the effect was more dramatic in the H2052 and H2452 cell lines (Figure 8, Appendix A). Of the four pro-survival genes, survivin and/or Mcl-1 levels were dramatically reduced in response to auranofin and palbociclib treatment. Bcl-xL loss occurred in all three cell lines with auranofin treatment, while palbociclib only induced Bcl-xL loss in the H2052 and H2452 cells. XIAP (X-linked inhibitor of apoptosis) expression remained unchanged by either drug.

## 3. Materials and Methods

### 3.1. Cell Lines and Reagents

The mesothelioma cell lines used in this study (H2052, H2373, H2452, and H2052) and the control cell lines, Met5A (we note as Met5A AD) and 3T3, were obtained from either the ATCC (American Type Culture Collection) or in collaboration with Dr. Robert Kratzke (University of Minnesota). All mesothelioma cell lines lack functional p16INK4a. The cells were grown in RPMI-1640 medium (Gibco BRL, Grand Island, NY, USA) supplemented with 10% newborn bovine serum (Sigma, St. Louis, MO, USA), and 1x concentration of antibiotic/antimycotic reagent (Gibco BRL, Grand Island, NY, USA) at 37 °C and 5% CO_2_. The following antibodies were utilized (Actin, Bax, Bcl-xL, Bid, cdc2 (CDK1), CDK2, CDK4, CDK6, Cyclin A, Cyclin B1, Cyclin D1, Cyclin E1, FOXM1, Mcl-1, pRb (S780), Rb, Survivin, and XIAP (see Appendix A for manufacturer and lot numbers). The CDK4/6 inhibitor, palbociclib, and auranofin were obtained from MedChemExpress (Monmouth Junction, NJ, USA).

### 3.2. Cell Proliferation Assays

Live cells, as determined by trypan blue dye exclusion assay, were counted on a Countess II FL Automated Cell Counter and plated on 96-well plates (3000 cells/100ul/well) in the RPMI media (including supplements). After 24 h, palbociclib or auranofin at varying concentrations was added to the wells and gently vortexed to mix. After a 72 h incubation at 37 °C and 5% CO_2_, 10 μL of solution from Cell Counting Kit–8 (Dojindo Laboratories, Kunamoto, Japan) was added to each well and the plates were incubated for 2 h. The plates were read at 450 nm using a SpectraMax M5 microplate reader (Molecular Devices, Sunnyvale, CA, USA). The reduction in optical density represented the reduction in mitochondrial succinate dehydrogenase activity, hence the reduction in surviving cell numbers. For drug combination studies, increasing amounts of one drug were added to H2373 or H2452 cells (0.5, 1, 2.5, 5, and 10 μM) while holding the other at a fixed concentration. Fixed concentrations were at the same 5 different concentrations (0.5, 1, 2.5, 5, and 10 μM). Combination studies were analyzed via the methods of Chou–Talalay, and Combination Indices (CI) were calculated via Compusyn [30]. In addition, ANOVA for multiple comparisons were utilized to evaluate for statistically significant differences amongst multiple groups. Experiments were conducted in triplicate. 

### 3.3. Colony-Forming Assays

Cells were grown in 6-well plates for 24 h, then treated with palbociclib or auranofin for 72 h [35]. The cells were then trypsinized, counted (including all dead cells) and plated on 24-well plates (400–1000cells/well). Plates were then grown for 10–14 days allowing single live cells to grow into colonies. Colonies were then fixed and stained with crystal violet and then read on the Licor CLx at 700 nm wavelength. The intensity of the colony staining was quantified using the Licor and associated software (Image Studio Version 5.2). 

### 3.4. Cell Cycle Analysis

After treatment with drug or dimethyl sulfoxide (DMSO) control for 48 h, mesothelioma cell lines were fixed in 70% cold ethanol by dropwise addition, and then stained with FxCycle^TM^ PI/RNase Staining Solution (Catalog #F10797, Invitrogen, USA), following the manufacturer’s protocol. Flow cytometry was performed using a FACSAria III Cell Sorter (BD Biosciences, San Jose, CA, USA), and results were analyzed by FACSDiva software Version 6.1.3 (BD Biosciences).

### 3.5. Immunoblotting

Cells were plated to reach a density of ~40% 24 h after plating and treated with the indicated concentration of drugs 24 h after plating. At 72 h of treatment, cells were harvested by scraping, and following centrifugation and washing with 1X PBS, cell pellets were flash frozen in powdered dry ice and stored at –80 °C. Cells were lysed with RIPA buffer (Harlow and Lane, CSH protocol) containing 1X Halt^TM^ Protease and Phosphatase inhibitor (Thermo Scientific, Waltham, MA, USA) following resuspension; lysates were flash frozen in powdered dry ice, thawed on ice, vortexed for 10 s, and centrifuged at 13,000 rpm at 4 °C for 10 min. The supernatant (lysate) was aliquoted, flash frozen in powdered dry ice, and stored at −80 °C. Protein content was determined using a Millipore (Burlington, MA, USA) Direct Detect Spectrometer. Twenty micrograms of protein were mixed with LI-COR (Lincoln, NE, USA) Loading Buffer (catalog 928–40004) containing 0.05 M DTT and heated at 95 °C for 5 min. Samples were subjected to electrophoresis on 4–20% Bio-Rad (Hercules, CA) Criterion TGX pre-cast polyacrylamide gels (catalog 5671094) run with Tris-Glycine Buffer (Bio-Rad, Hercules, CA, USA, catalog 1610772) and transferred to 0.2 µM Protran nitrocellulose membranes (GE healthcare Life Sciences, Marlborough, MA, USA) using Novex Transfer Buffer (Thermo Scientific, Waltham, MA, USA). Membranes were rinsed in Tris-buffered saline (Bio-Rad, Hercules, CA, USA, catalog 1706435) with 0.05% Tween 20 (Bio-Rad, Hercules, CA, USA, catalog 1610781) (TBS-T) and incubated in blocking buffer (5% bovine serum albumin (Sigma, St. Louis, MO, USA, catalog A9647) (BSA) in TBS-T) for 1 h at room temperature (RT). Blots were then incubated in primary antibodies diluted in TBS-T plus BSA overnight at 4 °C. The blots were washed with TBS-T, then incubated with secondary antibody, diluted in TBS-T plus BSA for 1 h at room temperature, and washed in TBS-T. Blots were developed with a chemiluminescence reagent (Pierce ECL PICO catalog 34,077 or DURA catalog 34,075 Western Blotting Substrate, Thermo Scientific, Rockford, IL, USA). Images were captured and analyzed with LI-COR (Lincoln, NE, USA) Odyssey FC System and Image Studio (version 5.2).

## 4. Discussion

Mitochondrial antioxidant defense and the cell cycle are two areas that may be targeted in mesothelioma. Previously, our group demonstrated via analysis of The Cancer Genomic Atlas that some tumors have increased mRNA expression associated with genes involved in antioxidant defense, such as Trx2 and Prx3 [36]. Previously, Cunniff et al. have demonstrated that inhibition of Prx3 by thiostrepton increases mitochondrial H_2_O_2_, disrupts mitochondrial energetics, and is associated with cytotoxicity in mesothelioma cells [12]. In addition, they demonstrated that thiostrepton and gentian violet, which downregulate TXN2, were associated with decreased tumor growth in mice mesothelioma xenografts [12]. We chose to evaluate auranofin, as it is an FDA-approved drug for rheumatoid arthritis and is known to downregulate Txn2 [19]. In our work, we demonstrated that it has clinically relevant potency against mesothelioma cells in vitro in cell proliferation and colon-forming assays. 

The tumor suppressor p16INK4a has been shown to be downregulated in up to 90–100% of mesothelioma tumors, depending on the histology (epithelioid, sarcomatoid, or biphasic) [25]. p16INK4a inhibits CDK4 and CDK6 (CDK4/6), which leads to cell cycle arrest [20,21,22,23,24]. The drug palbociclib is a selective CDK4/6 inhibitor and is one of three such inhibitors (along with abemaciclib and ribociclib) approved for use in advanced breast cancer, and, as such, are considered standard of care. However, up to now, there has been very little evaluation of CDK4/6 inhibition in mesothelioma. Two exceptions are the reports by Bonelli et al., where they evaluate palbociclib with or without PI3K/MTOR inhibition in mesothelioma cells [28,29]. They reported EC_50_ values ranging from 0.28 to 1.2 μM, which are slightly lower than what we observed. Part of this difference could be a slight variation in the protocols for cell proliferation assays. However, we also demonstrated that palbociclib results in a decrease in colony formation, which is a complementary technique to assess growth inhibition characteristics of a particular drug candidate. 

We subsequently conducted combination inhibitor studies of auranofin and palbociclib. CDK4 has been demonstrated to phosphorylate manganese superoxide dismutase (MnSOD), which is involved in superoxide processing to H_2_O_2_ [18]. In addition, CDK4 enhances activity of FOXM1, which has been demonstrated to affect mitochondrial antioxidant protein expression [17]. For several dosing combinations, we demonstrated that the combination of auranofin and palbociclib exhibited synergy against mesothelioma cells in proliferation experiments. 

The immunoblotting data suggest that in response to palbociclib inhibition of CDK4/6 activity, the loss of Cyclin A and B expression as well as CDK1 is consistent with cell cycle arrest. However, the increased expression of Cyclins D1 and E1 as well as CDK2, CDK4, and CDK6 suggests that compensation for CDK4/6 inhibition is being induced. Although auranofin also resulted in the loss of Cyclins A and B and CDK1 in H2052 and H2452, the effect was less pronounced, suggesting reduced cell cycle progression rather than arrest. This supposition is further supported by the cell cycle analysis data presented in Figure 6 and the observation that the palbociclib-treated cells used in the study showed substantial reduction in number relative to the untreated controls. Further, the enhanced expression of Cyclin D1 and CDKs 4 and 6 in response to palbociclib inhibition of CDK4/6 activity suggests that this is one possible mechanism by which the drug would lose effectiveness. Further, the dramatic upregulation of CDK2 activity and its partner Cyclin E is another mechanism that has been shown to promote resistance both acquired and intrinsic to CDK4/6 inhibitors [37,38].

In contrast, treatment by auranofin resulted in a gene expression profile that suggested overall loss of viability by multiple mechanisms, not just a predominant effect on cell cycle. Its mechanism of action (inhibition of cytosolic and mitochondrial thioredoxin reductase activity) would be expected primarily to induce ROS-driven cell death; however, it caused a substantial decrease in Cyclins A, B1, and D1, as well as all the CDKs and the cell cycle transcription factors FOXM1 and Rb, suggesting altered cell cycling as a contributing factor to loss of cell viability. However, in contrast to palbociclib, where arrest was induced at the G1/S transition, auranofin appears to be associated with a dramatic decrease in the percentage of cells at G1/S or G2/M checkpoints. Various studies have shown a significant role of mitochondrial-produced ROS in regulating the activation or degradation of the G1/S phase Cyclins D1, A, and E and G2/M Cyclin B1, implying that this is one potential method by which the reduced expression of the cyclins following auranofin treatment is occurring [39,40]. In one study of SOD1 inhibition with the compound called LD100 in HeLa and DU145 cells, exposure to LD100 resulted in reduced mRNA levels of CDK4, Cyclin D1, Cyclin E1, and Cyclin B1 and increased mRNA levels of CDKN1A and CDKN2D [41,42].

Transcriptional downregulation of the cyclins and CDKs due to loss of FOXM1 is another mechanism, as they are all know targets of FOXM1 [17]. However, the loss of FOXM1 was even more pronounced in the palbociclib-treated cells, yet Cyclin D1 and CDK expression was decreased only in the auranofin-treated cells. This is intriguing as it suggests the efficacy of the combination of the two drugs, with enhanced lethality at lower concentrations while in combination, which may be due to the inability of the cells in the presence of auranofin to upregulate the expression of the cyclins and CDKs to overcome the effects of CDK4/6 inhibition. 

Induction of pro-apoptotic Bax and Bid, both associated with mitochondrial-driven cell death, is consistent with a potential ROS mechanism. Inhibition of CDK4/Cyclin D1 kinase activation of SOD2 in the mitochondria by palbociclib and direct inhibition of thioredoxin 2 reduction in the mitochondria, which drives the detoxification of mitochondrial-generated ROS, are potential pathways for the increased expression of Bax and Bid. Loss of pro-survival gene expression was observed in all of the cell lines, with survivin being the only one that was consistent across all the mesothelioma cell lines in response to either drug. Mcl-1 and Bcl-xL loss was more consistently observed in the auranofin-treated cells, driving cell death as would be expected with the loss of redox capacity of both the mitochondrial and cytosolic thioredoxin systems. Taken together, these data suggest that cell death is the predominate mechanism resulting in loss of viability in auranofin-treated cells, which is consistent with its lethality at 2.5 µM in two of the three mesothelioma cell lines.

## 5. Conclusions

In conclusion, based on the constellation of findings presented, including the potent effects of palbociclib and auranofin against mesothelioma cell proliferation, colony formation, and the cell cycle, as well as the effect on Cyclin/CDK and pro-apoptotic/anti-apoptotic protein expression, we propose a potential model for how mitochondrial antioxidant defense and CDK4/6 may interact. Combination therapy with inhibition of mitochondrial antioxidant defense and CDK4/6 holds promise as a potential therapeutic strategy in mesothelioma. Future studies will be needed to fill in gaps and add to the model.

## Figures and Tables

**Figure 1 molecules-28-04380-f001:**
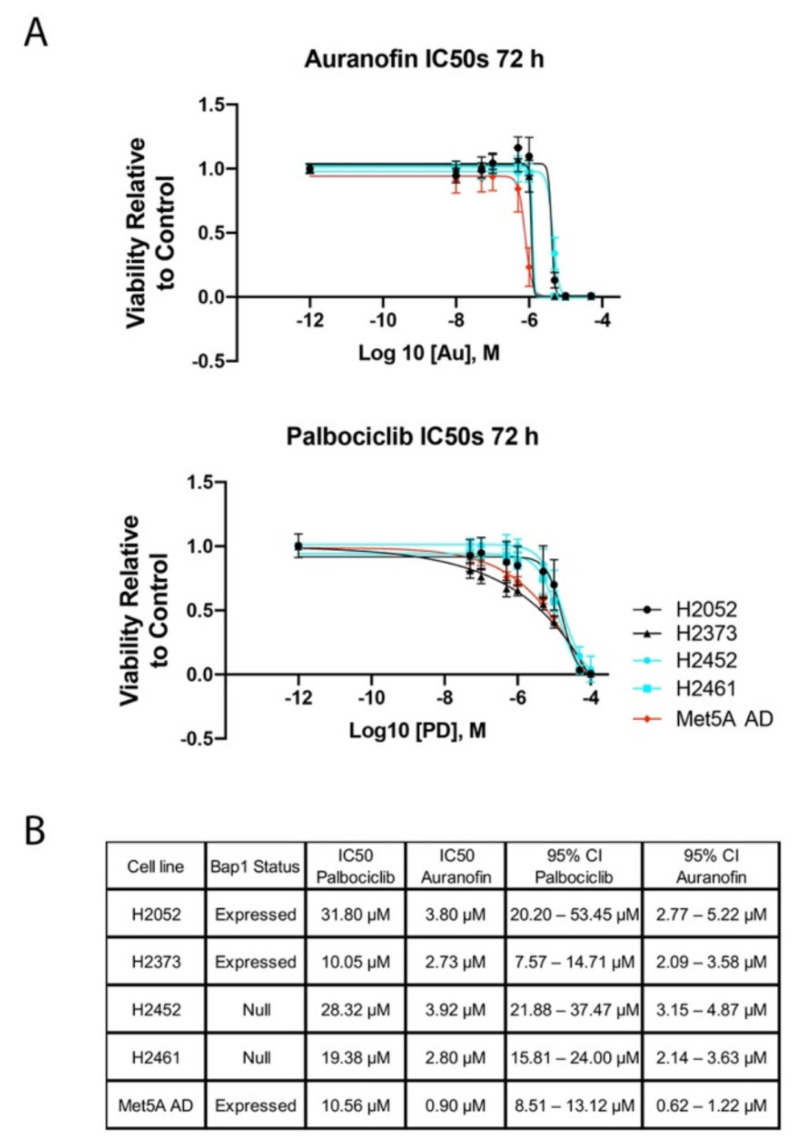
Determination of IC_50_ auranofin and palbociclib in mesothelioma cell lines. (**A**) Mesothelioma cell lines H2052, H2373, H2452, and H2461 were exposed to increasing concentrations of auranofin and palbociclib for 72 h. Cell viability was measured by CCK8 assay. (**B**) Table of IC_50_ of palbociclib and auranofin in mesothelioma cell lines.

**Figure 2 molecules-28-04380-f002:**
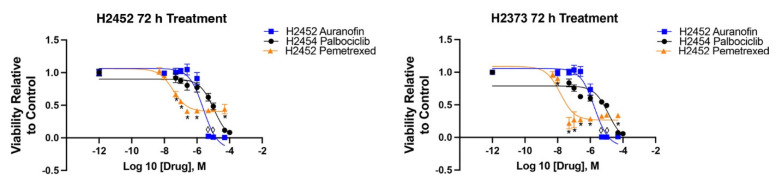
Determination of IC_50_ pemetrexed in mesothelioma cell lines. Mesothelioma cell lines H2373 and H2452 were exposed to increasing concentrations of pemetrexed for 72 h. Cell viability was measured by CCK8 assay. * Notes statistically significance.

**Figure 3 molecules-28-04380-f003:**
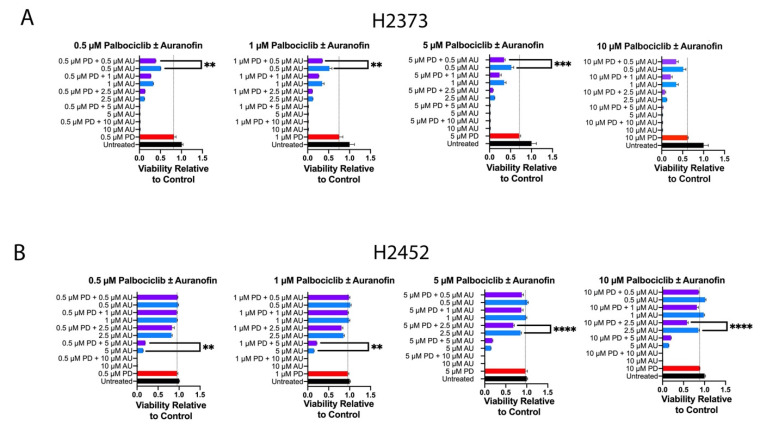
Combination studies with auranofin and palbociclib. (**A**) H2373 and (**B**) H2452 mesothelioma cell lines were incubated with combinations of palbociclib with auranofin at the indicated concentrations. Results of CCK8 assay are expressed as viability relative to control for each treatment. **, ***, **** denotes a statistically significant difference between the combinations, signifying a synergistic effect of the combination over either drug administered individually.

**Figure 4 molecules-28-04380-f004:**
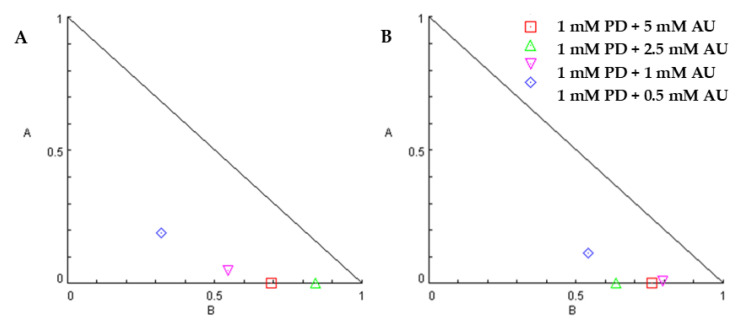
Isobolograms for auranofin and palbociclib. (**A**) H2373 and H2461 mesothelioma cell lines were incubated with combinations of palbociclib with auranofin at the indicated concentrations. (**A**) Isobologram for palbociclib plus auranofin in H2373 cells. (**B**) Isobologram for palbociclib plus auranofin in H2373 cells. Symbols denote palbociclib plus auranofin concentrations with CI < 1. In both figures, the axes A and B refer to the relative concentrations of drug A or drug B, and each individual point refers to a specific combination index value.

**Figure 5 molecules-28-04380-f005:**
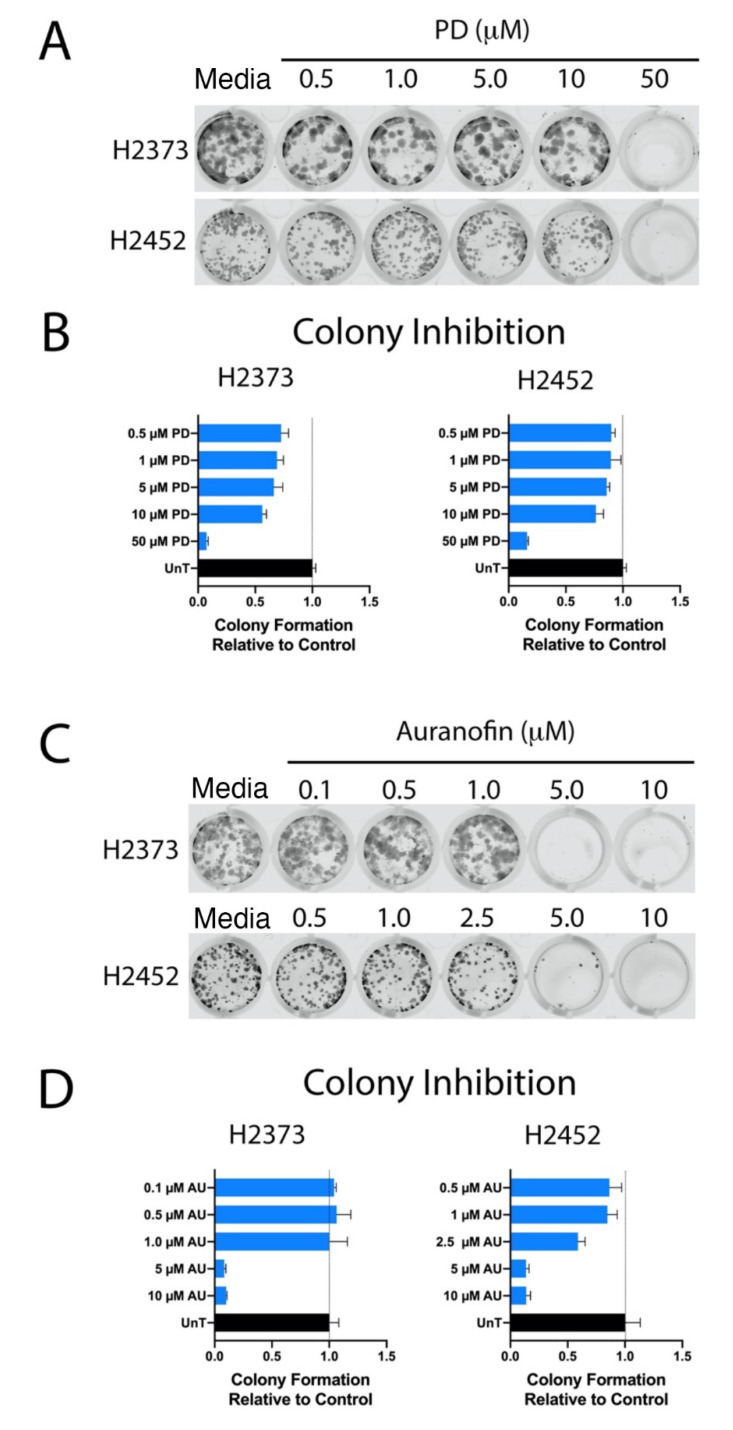
Inhibition of colony formation by palbociclib and auranofin. (**A**,**C**) H2373 and H2452 mesothelioma cell lines were treated with palbociclib (PD) and auranofin as single agents at the indicated concentrations and inhibition of colony formation was assessed. Cells were stained with crystal violet. Quantitation of relative inhibition of colony formation by (**B**) palbociclib and (**D**) auranofin compared to untreated controls. (*p* < 0.0001).

**Figure 6 molecules-28-04380-f006:**
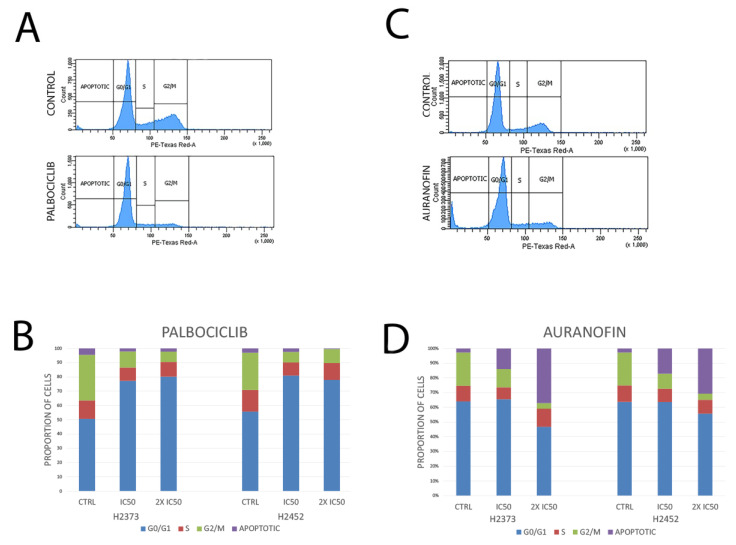
Effects of palbociclib and auranofin on the cell cycle of mesothelioma cell lines. Cell cycle analysis of H2373 mesothelioma cells treated with (**A**) palbociclib at IC_50_ concentration and at 10 μM and (**C**) auranofin at the IC_50_ concentration of 2.73 μM for 48 h. Palbociclib treatment resulted in increased accumulation of cells in the G1 phase. Auranofin resulted in an increased proportion of a sub-G1 population which is likely apoptotic. H2373 and H2452 cells were individually treated with (**B**) palbociclib and (**D**) auranofin at the measured IC_50_ concentration of the drug and 2 times the IC_50_ concentration of the drug for 48 h. Flow cytometric analysis of the cell cycle was performed and the proportion of cells in each phase of the cell cycle is indicated. Data represent an average of two independent experiments.

**Figure 7 molecules-28-04380-f007:**
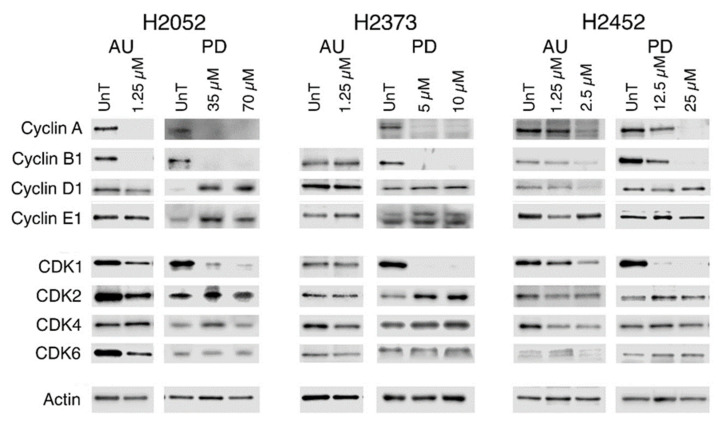
Expression of cyclins and CDKs in auranofin- or palbociclib-treated mesothelioma cell lines. Representative Western blots show the expression pattern and levels of the cyclins and CDKS from 2 to 3 independent experiments. No Cyclin A signal was detected for the H2373 auranofin-treated cells despite repeated attempts. Actin was used as the lane loading control and is shown at the bottom and represents the signal observed for the Western blots used for the figure. The cell lines and treatments are indicated above the blots. The proteins examined are indicated on the left. AU, auranofin; PD, palbociclib.

**Figure 8 molecules-28-04380-f008:**
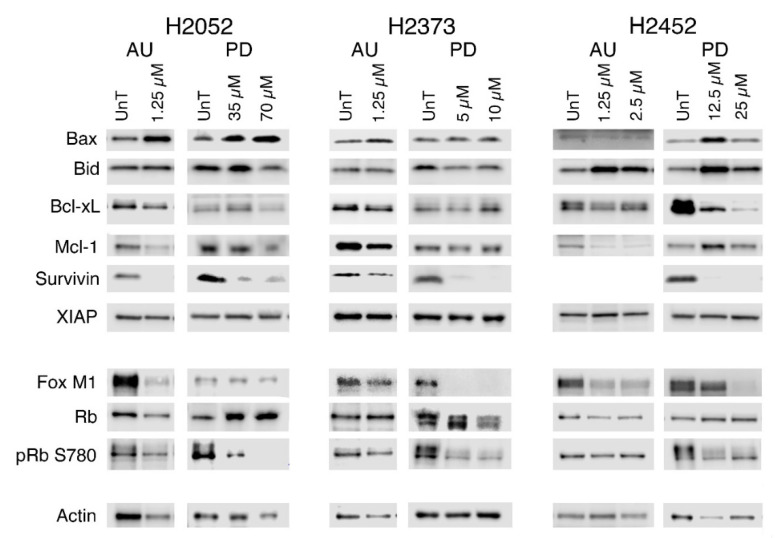
Expression of survival genes and cell cycle transcription factors in auranofin- or palbociclib-treated mesothelioma cell lines. Western blots showing representative expression patterns of pro-apoptotic (Bax and Bid) and pro-survival (Bcl-xL, Mcl-1, survivin, and XIAP) following 72 h of treatment from 2 or 3 independent experiments. No survivin signal was detected in the auranofin-treated H2452 cell line despite repeated attempts and the signal in the palbociclib-treated cells required extended duration ECL for detection. Actin was used as the lane loading control and is shown at bottom and represents the signal observed for the Western blots used for the figure. The cell lines and treatments are indicated above the blots. The proteins examined are indicated on the left. AU, auranofin; PD, palbociclib.

**Table 1 molecules-28-04380-t001:** Effect size (percent inhibition in decimals) with combination indices (CI).

Effect Size (Percent Inhibition in Decimals) with Combination Indices (CI)									
Dose PD (μM)	Dose AU (μM)	H2373		H2452		H2052		H2461	
		Effect	CI	Effect	CI	Effect	CI	Effect	CI
0.99	1.39	0.99	1.42	0.99	4.24 × 10^14^	0.97	1.25	0.99	1.52
0.99	0.69	0.99	0.71	0.79	4.65 × 10^8^	0.93	0.82	0.99	0.76
0.89	0.76	0.89	0.74	0.4	3.26 × 10^5^	0.55	1.11	0.97	0.54
0.63	0.69	0.63	0.67	0.13	6.14 × 10^2^	0.05	3367.61	0.59	0.58
0.48	1.30	0.48	1.32	0.08	6.37 × 10^1^	0.01	502,919.00	0.45	0.37
0.99	1.39	0.99	1.42	0.99	2.12 x 10^14^	0.97	1.25	0.99	1.52
0.99	0.69	0.99	0.71	0.82	5.19 x 10^8^	0.94	0.78	0.99	0.76
0.87	0.81	0.87	0.79	0.23	5.61 × 10^3^	0.46	1.33	0.94	0.68
0.6	0.65	0.6	0.63	0.05	4.97	0.06	938.60	0.48	0.67
0.46	0.92	0.46	0.91	0.01	1.03	0.04	3419.45	0.33	0.46
0.99	1.39	0.99	1.42	0.99	4.24 × 10^13^	0.96	1.37	0.99	1.52
0.99	0.69	0.99	0.71	0.82	1.04 × 10^8^	0.92	0.86	0.99	0.76
0.85	0.85	0.85	0.82	0.18	3.10 × 10^2^	0.42	1.14	0.95	0.64
0.55	0.59	0.55	0.56	0.05	1.89	0.09	50.45	0.34	0.80
0.42	0.51	0.42	0.49	0.02	0.81	0.02	5958.66	0.16	0.66
0.99	1.39	0.99	1.42	0.99	2.12 × 10^13^	0.96	1.37	0.99	1.52
0.98	0.86	0.98	0.87	0.83	6.94 × 10 ^7^	0.9	0.93	0.99	0.76
0.83	0.89	0.83	0.86	0.19	2.05 × 10^2^	0.37	1.20	0.94	0.68
0.49	0.63	0.49	0.60	0.05	1.51	0.01	25,147.50	0.35	0.79
0.28	0.87	0.28	0.84	0.03	0.72	0.01	25,146.70	0.06	1.65
0.99	1.39	0.99	1.42	0.99	4.24 × 10^12^	0.97	1.25	0.99	1.52
0.99	0.69	0.99	0.71	0.88	7.65 × 10^7^	0.91	0.90	0.99	0.76
0.8	0.94	0.8	0.91	0.3	5.12 × 10^2^	0.47	0.96	0.95	0.64
0.36	0.73	0.36	0.69	0.16	17.7	0.01	5030.72	0.43	0.71
0.14	1.45	0.14	1.41	0.13	6.52	0.01	5029.95	0.15	0.57

## Data Availability

Please contact M.A.K. for inquiries.

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
