# Peer review of "Inhibition of Mitochondrial Antioxidant Defense and CDK4/6 in Mesothelioma"

_molecules, 2023, doi:10.3390/molecules28114380_

Round 1

Reviewer 1 Report

This manuscript addresses the use of two clinically used drugs, the mitochondrial antioxidant defense inhibitor auranofin, used as an antirheumatic agent, and cyclin-dependent kinase 4/6 inhibitor Palbociclib, used to treat HR-positive and HER2-negative breast cancer, alone or in combination in the treatment of mesothelioma. Mesothelioma is a highly aggressive type of cancer with bad prognosis and only 8% five-year survival rate. The group determined the effect of this drugs in several different assays as cell survival, colony formation, cell cycle progression and analyzed the expression of some proteins involved in these processes. Although I found a pleasant reading and with useful information, I think it cannot be accepted in its current form.

In my point of view the Manuscript need major review of its present data.

Major revision:

11)      In figure 1 the IC50 was determined for several mesothelioma cell lines and one non transformed cell. Although the selective index was not calculated it seems that there is no selectivity for these drugs. How do the authors propose a use for them in the treatment of this type of cancer if there is no therapeutic window? The drug combination increases the selectivity of the individual drug?  

22)      To determine if there is a synergic or additive effect of the drugs, I think is important to calculate the coefficient of drug interaction (CDI) of the cytotoxicity (IC50) of the isolated compounds and their combination. Without this information it is hard to conclude if there is in fact and synergic effect as discussed and concluded by the authors.

33)      No other experiment was done using the drug combination. How is the cyclin, CDK and apoptotic related proteins profile with the drug combination? Is there an additive or synergic effect in any of this protein expression that might correlate with the cytotoxicity experiment? Is there and synergic of additive in inhibition of colony formation or cell cycle control?

44)      There is a lack of comparation with other chemotherapeutic drugs as control.     

Author Response

Reviewer 1

Major revision:

11)      In figure 1 the IC50 was determined for several mesothelioma cell lines and one non transformed cell. Although the selective index was not calculated it seems that there is no selectivity for these drugs. How do the authors propose a use for them in the treatment of this type of cancer if there is no therapeutic window? The drug combination increases the selectivity of the individual drug? 

Response: These are excellent questions. However, based on prior experiments in other labs and in this manuscript, we hypothesize that the answer will be ultimately much more complex than can be addressed here. Auranofin and palbociclib are both drugs that have been FDA-approved and in clinical practice for several years. The IC50 values for auranofin and palbociclib were similar in 3T3 cells and mesothelioma cells. However, we know that each drug is rather well tolerated by patients. It is possible that there is a significant variation in how normal cells are affected in patients versus in vitro models, as 3T3 cells are one isolated representative control cell line.

22)      To determine if there is a synergic or additive effect of the drugs, I think is important to calculate the coefficient of drug interaction (CDI) of the cytotoxicity (IC50) of the isolated compounds and their combination. Without this information it is hard to conclude if there is in fact and synergic effect as discussed and concluded by the authors.

Response: In addition to the figure showing the drug combinations in the original manuscript, we added Combination Indices per the methods of Chou and Talalay.

33)      No other experiment was done using the drug combination. How is the cyclin, CDK and apoptotic related proteins profile with the drug combination? Is there an additive or synergic effect in any of this protein expression that might correlate with the cytotoxicity experiment? Is there and synergic of additive in inhibition of colony formation or cell cycle control?

Response: We did not conduct these experiments originally, as we were not sure there would be synergy. There was not enough time to repeat all these experiments. We anticipate conducting these and reporting in a subsequent manuscript.

44)      There is a lack of comparation with other chemotherapeutic drugs as control.     

Response: Currently, cisplatin, pemetrexed, nivolumab, and ipilimumab are options for standard of care systemic treatment in mesothelioma. All have been extensively studied in other labs prior to FDA approval. We conducted new experiments with pemetrexed for control and have included that in the revised manuscript.

Reviewer 2 Report

Comments for the authors:

I reviewed the paper "Inhibition of Mitochondrial Antioxidant Defense and CDK4/6 in Mesothelioma", which should be a valuable work from the level of molecular mechanism to provide a combined drug administration strategy for the treatment of mesothelioma. I found it can be accepted after major revision and some comments should be done to improve the article. The following corrections should be made.

1. Authors should elaborate on why mesothelioma is difficult to treat and cite relevant literature. The author's summary of mitochondrial antioxidant defense is not comprehensive.

2. In figure 2, authors should add statistical analyses of the combined administration group and the PD treatment group. Here, the number of parallel trials and a description of statistical significance analysis should be added. Figure 3 also lacks statistical significance analysis and related descriptions.

3. In figure 5, AU treated H2452 cells should be retested for the CDK6 characteristic, and two bands in close proximity should be unreasonable. The same problem exists in the cyclin E1 characteristic of PD treated H2373 cells, and the Rb characteristic of PD treated H2373 cells.

4. The results of the WB test should be given for proteins with no bands detected.

5. Authors should provide quantitative results of all WB tests, and it is suggested to summarize them in table form. Authors can refer to https://doi.org/10.1155/2019/9706792

6. The overall conclusion needs to be improved. Use regular articles as: https://doi.org/10.1155/2019/9706792, https://doi.org/10.3390/ molecules27010148, https://doi.org/10.1038/s41568-021-00435-0, https://doi.org/10.1038/s41580-021-00404-3.

Author Response

Reviewer 2

  1. Authors should elaborate on why mesothelioma is difficult to treat and cite relevant literature. The author's summary of mitochondrial antioxidant defense is not comprehensive.

Response: We have cited the relevant literature for standard of care mesothelioma treatment in the advanced setting. In addition, we added over a paragraph summary of mitochondrial antioxidant defense.

  1. In figure 2, authors should add statistical analyses of the combined administration group and the PD treatment group. Here, the number of parallel trials and a description of statistical significance analysis should be added. Figure 3 also lacks statistical significance analysis and related descriptions.

Response: For the experiments shown in figure 2, we have included in the methods section the use of ANOVA for comparison amongst multiple groups and the fact that all experiments were conducted in triplicate.

  1. In figure 5, AU treated H2452 cells should be retested for the CDK6 characteristic, and two bands in close proximity should be unreasonable. The same problem exists in the cyclin E1 characteristic of PD treated H2373 cells, and the Rb characteristic of PD treated H2373 cells.

Response: We believe this to be a characteristic of potential overexposure to detect these bands.  To get good exposure for the CDK6 band it overdevelops the CDK4 band.

  1. The results of the WB test should be given for proteins with no bands detected.

Response: We have included the quantitation for all bands in the Western blots.

  1. Authors should provide quantitative results of all WB tests, and it is suggested to summarize them in table form. Authors can refer to https://doi.org/10.1155/2019/9706792

Response: We have included the relative quantitation of bands for all Western blots. Due to the large amount of data, we included this as a supplementary table.

  1. The overall conclusion needs to be improved. Use regular articles as: https://doi.org/10.1155/2019/9706792, https://doi.org/10.3390/ molecules27010148, https://doi.org/10.1038/s41568-021-00435-0, https://doi.org/10.1038/s41580-021-00404-3.

Response: We incorporated additional articles to strengthen the conclusion.

Round 2

Reviewer 1 Report

The manuscript has improved since the last revision. 

Reviewer 2 Report

The authors have answered almost all of my questions head-on. I think it can meet the requirements of journal publication on the whole.